# Control Schemes for Room-Scale VR Games

Author 1*

Organization or School

## ABSTRACT

Since the 1990's, most desktop 3D games have adopted the "mouselook" control scheme, in which the mouse simultaneously rotates the camera view, aims at targets and steers the avatar. Control schemes for virtual reality games are less standardized and must integrate input from additional devices into the control scheme, namely the head-mounted display and position sensors. We conducted a mixed-methods study to evaluate the usability of two common control schemes in VR games. The first was coupled (or gaze-directed, where the player moves in the direction the camera faces), and the second was decoupled (or hand-directed, where the player moves in the direction the avatar faces while being free to look around without affecting movement direction). Our participants used an Oculus Rift CV1 head-mounted display, Oculus Touch motion controllers and two positional sensors as input devices. We did not find significant differences between the control schemes in terms of quantitative usability metrics. However, our qualitative results indicated usability issues with the decoupled control scheme. When using the decoupled control scheme, participants found it difficult to maintain awareness of their avatar's facing direction. The coupled control scheme was the most usable as judged by consistently positive feedback and the absence of major usability issues. These results highlight the importance of gathering qualitative user feedback in addition to quantitative usability metrics.

**Keywords**: Virtual Reality Games, Room-Scale Virtual Reality, Control Schemes, Control Mappings, Head-Mounted Displays, Player Experience, Locomotion, Travel Techniques.

**Index Terms**: •Human-centered computing~Human computer interaction (HCI)~Interaction paradigms~Virtual reality•Software and its engineering~Software organization and properties~Contextual software domains~Virtual worlds software~Interactive games•Human-centered computing~Human computer interaction (HCI)~HCI design and evaluation methods~Usability testing

## 1 INTRODUCTION

Virtual reality (VR) is rapidly gaining popularity as a gaming platform, prompted by advances in consumer-grade head-mounted displays [Epp et al. 2021]. Modern room-scale VR devices, such as Oculus' Oculus Rift and Quest and HTC's Vive, use various sensors to track the user. The head-mounted display (HMD) tracks head orientation and handheld motion controllers track hand movements. The sensors (either internal or outward-facing) capture hand and head position within a tracked space. It is possible to map these device inputs to several in-game functions, such as rotating the camera view, rotating the player's avatar and aiming at targets—creating a myriad of control mapping

combinations, or control schemes. The dominant control scheme for desktop 3D games is called "mouselook" [Cummings 2007], in which moving the mouse simultaneously rotates the player avatar and camera view and the aiming target reticle is fixed to the center of the screen. However, VR Game designers must integrate additional input modalities into a game's control scheme. A variety of control schemes have been implemented in commercial VR games [Al Zayer et al. 2020; Di Luca et al. 2021], but no research has evaluated their usability for room-scale VR games.

To address this gap, we evaluated control schemes for room-scale, controller-based locomotion in a first-person shooter (FPS) game. We first identified two common control scheme types: coupled and decoupled. In coupled control schemes, the player always moves in the direction the camera is facing. In decoupled schemes, the player can look around the environment freely without affecting the direction of movement, as camera and movement direction are "decoupled". Previous research has found that coupled control schemes yield superior performance in both non-gaming [Ruddle et al. 2013a; Bowman et al. 1997; Bowman 1999] and gaming contexts [Martel et al. 2015; Martel and Muldner 2017]. However, these studies did not use modern room-scale VR gaming systems that track player movement and blend real-walking with controller-based locomotion.

We conducted a study to compare the usability of coupled and decoupled control schemes using the Oculus Rift CV1 [Oculus VR 2013] head-mounted display, Oculus Touch motion controls and room-scale position tracking for the commercial game Serious Sam VR: The First Encounter (SSVR:TFI) [Croteam 2017a]. As the effectiveness of an interaction technique is influenced by the task context [Abeele et al. 2013; Lampton et al. 1995; Mine 1995; Ware and Osborn 1990], we evaluated control schemes using two tasks. The first task was an ecologically valid gaming context task where participants played a first-person shooter game level. The second task was a single path maze task we developed to study the effect of control scheme on locomotion.

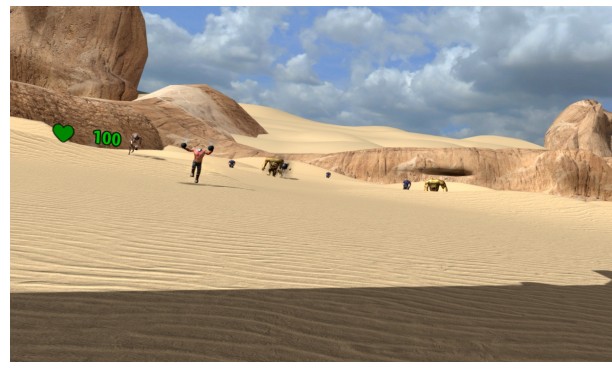

Figure 1: Serious Sam VR: The First Encounter, a VR first-person shooter game by Croteam.

---

* email address

The study employed a mixed-methods approach, with quantitative usability metrics and also gathering and analyzing qualitative feedback in order to probe for reasons behind the quantitative results. Our goal was to identify factors affecting control scheme usability and game immersion and to provide recommendations for designing usable FPS VR game control schemes. Our research questions included:

R1: Does the control scheme affect usability (effectiveness, efficiency, satisfaction and learnability)? Hypothesis: The coupled control scheme will be more usable, based on prior VR research [Bowman et al. 1999; Mine 1995; Ruddle et al. 2013b; Martel and Muldner 2017; Suma et al. 2007].

R2: Does the control scheme affect game immersion? Hypothesis: The decoupled scheme will produce higher immersion scores. Research in control schemes for desktop 3D games has revealed that head-tracking input is more immersive than gamepad or mouse + keyboard despite negatively effecting performance [Ilves et al. 2014; Kulshreshth and Laviola 2013; Sko et al. 2013; Sko and Gardner 2009; Wang et al. 2006], while game research for seated VR experiences has been inconclusive [Martel et al. 2015; Martel and Muldner 2017].

R3: Does the task context influence usability scores? Hypothesis: The coupled scheme will be more usable for both experimental tasks. Past research has shown that coupled schemes are more usable for locomotion tasks [Bowman et al. 1999; Mine 1995; Ruddle et al. 2013b; Martel and Muldner 2017; Suma et al. 2007]. VR control schemes in an ecologically valid gaming task.

R4: What attributes affect control scheme usability? Hypothesis: Schemes that are similar to existing game controls and thus more familiar to "gamers", such as the coupled scheme, will be more usable. [Martel et al. 2015; Martel and Muldner 2017].

## 2 RELATED WORK

A control scheme refers to the control-display mappings [MacKenzie 2013] within a given interaction technique. The term control scheme is primarily used by game developers, as games are intensely interactive and require a complex set of control-display mappings [Harris 2008]. For first-person desktop 3D games, control schemes allow the player to perform the following in-game actions:

- **Rotating the in-game camera view,** to look around the game world.

- **Rotating the avatar,** or "steering" to specify the movement direction through the game world.

- **Targeting objects in the environment**. Moving a selection cursor or targeting reticle to aim at enemies in a FPS game.

In VR contexts, game designers must decide how to integrate additional input devices, such as the HMD, handheld controllers, and position sensors into a control scheme. Most consumer-grade VR gaming systems support room-scale positional tracking and provide the player's head and hand pose as they move around the tracked environment. However, while these systems support real-walking for locomotion, the tracked area is usually smaller than the virtual environment—thus requiring other travel techniques to allow the player to move longer distances in the virtual environment. Current VR games have adopted a variety of travel techniques to supplement real-walking-based locomotion. Boletsis and Cedergren created a typology of the current VR locomotion techniques based on a literature review of techniques studied between 2014 and 2017 [Boletsis 2017]. We use the categories they defined for the various VR locomotion techniques.

In room-scale tracked spaces, continuous and non-continuous motion types allow players to travel the longer distances required by large virtual environments. Non-continuous locomotion techniques reduce cybersickness, as demonstrated in point-to-teleport [Frommel et al. 2017; Funk et al. 2019; Boletsis and Cedergren 2019; Prithul et al. 2021], translation snapping [Farmani and Teather 2018; Farmani and Teather 2018], dash locomotion and out-of-body locomotion [Griffin and Folmer 2019] techniques. This advantage makes non-continuous locomotion suitable for games in many genres [Di Luca et al. 2021] and for use in non-game VR applications. However, first-person shooter and other action game genres introduce unique usability requirements For example, in action genres such as FPS it is essential that players can maintain visual contact with threats in the game world. Non-continuous techniques make this more difficult by rapidly changing the user's point of view during teleporting, snapping or dashing. Furthermore, for multiplayer games, a consistent avatar position is important so that players can see and target enemies; teleportation actively breaks this game mechanic. Because of these issues, most VR FPS use continuous motion where the player's viewpoint moves continuously through the virtual world in response to joystick or other input device [11, 23]. The continuous motion type in Boletsis and Cedergren's taxonomy is associated with "controller-based" locomotion, often called "full locomotion". For these reasons, we will evaluate control schemes for controller-based locomotion in a room-scale VR FPS game.

For controller-based VR games, there are two main approaches to control scheme design: coupled schemes and decoupled schemes [Al Zayer et al. 2020; Di Luca et al. 2021]. In coupled control schemes, the user moves in the direction the camera is facing. decoupled schemes allow the player to look around the environment freely without affecting movement direction. In both coupled and decoupled schemes, targeting enemies or objects in the virtual environment is mapped to motion controller ray and decoupled from the camera view.

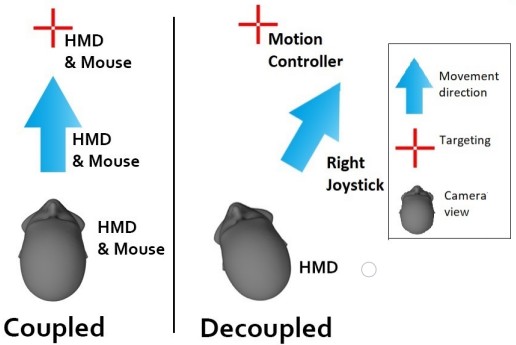

Figure 2: Visualizations of coupled, decoupled control schemes.

While many studies have compared the effectiveness of different locomotion techniques, few have evaluated the usability of control schemes for a locomotion technique. The few gaming studies that have evaluated control schemes [Martel and Muldner 2017; Martel et al. 2015], did so for seated VR, and used the mouse and keyboard as input devices rather than motion controllers. These studies also did not evaluate control schemes in the context of an ecologically-valid gaming task. Results from these studies suggested that coupled control schemes were preferred by participants, resulted in better player performance on targeting and locomotion tasks and fostered better game immersion. The decoupled schemes were considered difficult to use because their viewpoint did not move in direction the camera faced, which participants expected from using traditional mouselook control schemes; rather, they moved in the direction that the avatar faced. Players found that they could not maintain awareness of their avatar's facing direction separate from the camera direction. However, one study found that game immersion was highest with the decoupled scheme despite the negative effect the scheme had on performance scores [Martel et al. 2015]

## 2.1 VR Interaction Techniques

We now describe work on interaction techniques for virtual environments in non-gaming contexts. Bowman et al. devised a taxonomy of interaction techniques for common tasks in virtual environments [Bowman 1999]: travel, or locomotion ("the movement of the user's viewpoint from place to place"), selection ("indicating virtual objects within the environment"), and manipulation ("changing object properties such as position and orientation"). Bowman's travel taxonomy included "gaze-directed" direction selection (or "steering"), gesture steering, discrete selection, and 2D pointing. We evaluated control schemes that use the modern equivalent of two of these: (1) the gaze-directed technique, which we refer to as "coupled"; and (2) the 2D pointing technique, which we refer to "decoupled". Note that there are no commonly agreed upon names for these interaction techniques. Gaze-directed steering has also been referred to as "head-coupled", "view-directed" and "head-directed", while 2D pointing-steering has been referred to as "non-head-coupled", "hand-directed" and "decoupled".

VR researchers hypothesized that coupled travel would be most natural for VR applications. Bowman et al. conducted two experiments that compared "gaze-directed" travel, with steering using a 2D pointing device [Bowman et al. 1999]. The first experiment tested how fast participants could move to a target object and showed the effect of control scheme did not significantly affect travel time. The second experiment showed that coupled "gaze-directed" travel was significantly faster for moving to positions relative to a target object, for example, moving to a point in front of a target. Relative positioning is useful to gain a view of a target object, which could have applications in gaming to gain an advantageous position relative to enemies. Ruddle found that coupled travel that used head tracking to control movement direction (coupled or "gaze-directed" travel) was most effective as compared to other schemes [Ruddle et al. 2013b].

For targeting tasks, early VR research found that head tracking was more effective than joystick input. Lampton et al. found that head tracking was the most effective input for targeting [Lampton et al. 1995]. Pausch et al. obtained similar results in their study that compared HMD and joystick as input devices for locating targets in a virtual world [Pausch et al. 1993].

These early VR studies demonstrate the importance of task context when evaluating a control scheme. Coupled control schemes were more usable for travel tasks, while decoupled schemes aided targeting tasks. Nevertheless, the advances in virtual reality hardware and software require that these results are updated for modern contexts—and also for modern gamers, who are accustomed to desktop 3D games and have pre-existing control scheme biases. More recent, room-scale VR studies have evaluated control schemes for locomotion. For example, Suma et al. report that a coupled control scheme facilitated faster travel through a single-path maze compared to a decoupled control scheme but found no significant difference in the number of collisions with maze boundaries [Suma et al. 2007]. Christou and Aristidou compared coupled, decoupled and teleport locomotion techniques, finding that the decoupled control scheme produced significantly lower success rates than the coupled and teleport methods in a search task [Christou and Aristidou 2017]. Riecke et al. also found that a coupled control scheme performed as well as real walking on the majority of their DV in a search task [Riecke et al. 2010]. Thus, the coupled control scheme has been shown to be more usable for travel tasks, both in early and current VR research.

## 2.2 Game Immersion

Game immersion is "concerned with the specific, psychological experience of engaging with a computer game" [Jennett et al. 2008]. Researchers have identified several sources of game immersion. Mayra and Ermi's model includes challenge, sensory, and imaginative sources of immersion and was developed by analyzing interviews that asked participants to identify what elements held their interest when playing a game [Mäyrä and Ermi 2005]. Brown and Cairns used grounded theory to develop a model of immersion with control and challenge factors [Brown and Cairns 2004] and found that controls feel effortless when they balance of control and challenge. Jennet et al. developed a game immersion questionnaire based on the immersion factors identified in a literature review. They refined their questionnaire over the course of three experiments, which validated their immersion model and included five factors: cognitive involvement, emotional involvement, real-world dissociation, challenge and control [Jennett et al. 2008].

We will use the Immersive Experiences Questionnaire that Jennet et al. developed to measure game immersion for each control scheme in a room-scale VR game context. Research has demonstrated that VR games are more immersive than desktop 3D games [Lugrin et al. 2012; Lugrin et al. 2010; Tan et al. 2015; Yoon et al. 2010].

## 2.3 Related Work Summary

To date, the few studies that compared control schemes for VR games found the coupled control scheme more usable. However, prior work has not investigated the impact of different control schemes for room-scale VR games, only seated VR games. Research in non-gaming contexts also reported that coupled control schemes outperformed decoupled control schemes for locomotion, but targeting tasks were more suited to decoupled control schemes. Thus, it is not clear how a given control scheme affects the player experience for controller-based room-scale VR FPS games—which is needed to design VR games that are usable and immersive.

## 3 METHODOLOGY

### 3.1 Participants

We recruited 28 participants (aged 18 to 30, mean age of 20) through a university study recruiting system, posters and word-of-mouth. Two participants withdrew from the study due to cybersickness, so our analysis is based on 26 participants (19 male, aged 18 – 30, mean age of 20 years). Our Game Experiences Questionnaire asked participants to rate their skill in playing 3D games on a scale of 1 to 5. The mean skill level for the sample was 3.3. Of the 28 participants, 15 had used motion controllers for PC or console games, 19 had used a gamepad and 19 had used mouse and keyboard. One participant was an exclusively mobile gamer. Only 1 participant was a frequent VR gamer, 20 had never used VR and 8 had only tried it once or twice.

### 3.2 Apparatus

#### 3.2.1 Hardware

We used a gaming setup that included a desktop PC with Intel i7-7700, 4.20GHz CPU with 32 GB RAM and two NVIDIA GeForce 1080 video cards in a dual-card SLI setup. The VR gaming setup included Oculus Rift CV1, the Oculus Touch motion controllers and two external sensors. The controller buttons were mapped to in-game actions as labeled in Figure 3. The CV1 has two external sensors with a tracking volume of 100°H x 70°V. The sensor's tracking range is 10 feet. Our testing environment tracked an area of 8 x 8 feet.

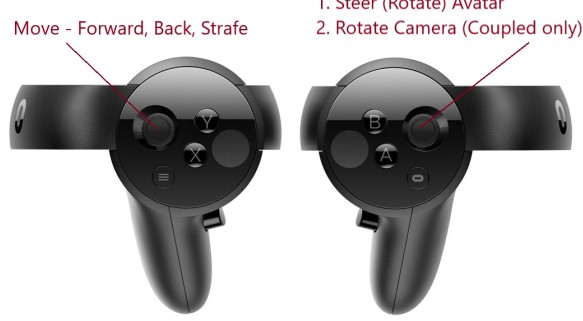

Figure 3: Oculus Touch control mappings.

#### 3.2.2 Software

Our study used the VR-enabled first-person shooter game Serious Sam: The First Encounter (SS:TFE) [Croteam 2017a]. We chose this game for two reasons: (1) it includes the Serious Sam Fusion Editor 2017 [Croteam 2017b], a level editing tool by Croteam that we used to develop custom testing environments, and (2) it supports both coupled and decoupled control schemes.

**Coupled Control Scheme.** The coupled control scheme in SS:TFE is called "head-oriented" in the options menu. With this control scheme, HMD input is coupled to the right joystick; both control the same functions, meaning the player can use either the right joystick or the HMD (or a combination of both) to rotate the camera viewpoint. This control scheme is similar to the "mouselook" control scheme used in desktop 3D games. However, in SS:TFE, targeting is achieved by raycasting from the Oculus Touch controller and is decoupled from the camera view. The player essentially "points" the motion controller toward the desired target.

**Decoupled Control Scheme.** The decoupled control scheme in SS:TFE is called "hand-oriented" in the options menu. Unlike coupled controls, this control scheme assigns different functions to the right joystick and the HMD. In this scheme, the right joystick controls steering, while the HMD controls the camera view, allowing the player to look around the environment independent from the avatar's movement direction. Targeting is achieved by raycasting from the Oculus Touch controller and is decoupled from camera in the same manner as the coupled control scheme.

#### 3.2.3 Game Maps

We developed two custom game maps using the *Serious Sam Fusion Editor 2017*. These custom maps were designed to evaluate control schemes in different task contexts.

**Gaming context map.** We designed the gaming context map to provide an ecologically valid test of the control schemes by measuring participants' performance as they played against AI non-player characters in the *Bends on Sand* level from SS:TFE, played in survival mode (see Figure 4). We instructed participants to shoot as many enemies as possible while avoiding the attacks of AI-controlled enemies. We recorded the number of kills and survival time.

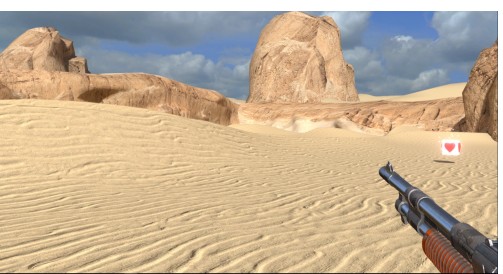

Figure 4: In-game view of the SS:TFE *Bends on Sand* level.

**Locomotion map.** A single-path maze, designed to test travel in complex environments. Participants move from the start point to end point while avoiding collision with maze walls. Custom scripting deducted one health point for each second the participant touched a wall. Participants had 100 Health points at the start of the task and each health point lost was counted as an "error". See Figure 5.

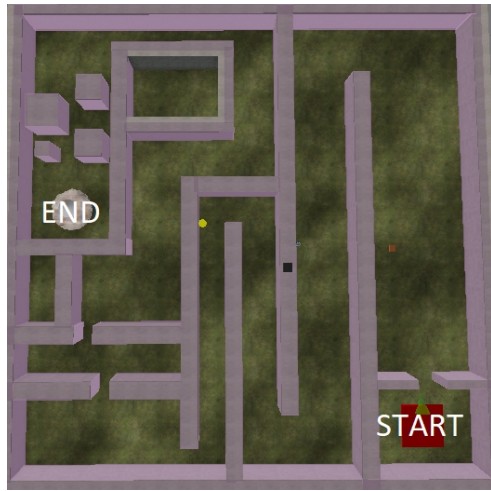

Figure 5: Top-down view of the locomotion map.

### 3.3  Procedure

We conducted the experiment in a lab equipped with hardware described in the Apparatus section. Upon arrival, we asked participants to provide informed consent. We then gave a brief verbal introduction to the study procedures and goals. Next, participants completed the Game Experience Questionnaire.

The first phase of the experiment allowed participants to acclimatize to VR. To begin, participants took a pre-test Cybersickness Rating Scale, where they rated their cybersickness on a scale of 0 to 10. Then they were fitted with the Oculus Rift CV1 headset and played the Karnak Temple level using the first control scheme (orders were counter-balanced) for a few minutes. After participants became acclimatized, we asked them to complete the Cybersickness Rating Scale again and then asked them if they were experiencing any cybersickness symptoms. At this point, two participants withdrew from the study.

In the next phase of the experiment, participants completed four trials of the game context map task using the Bend on Sand level, for each control schemes. We recorded each participant's performance scores (survival time, number of kills) for each condition and trial. After the game context task, participants completed the Immersive Experiences Questionnaire for each control scheme. There were two applications of the IEQ per test session, one per control scheme.

Next, participants completed one trial of the locomotion map task per control scheme. We recorded each participant's performance scores (time, number of errors) for each condition. Participants completed the Task-Specific Control Scheme Questionnaire (TCSQ) after using each scheme. There were two applications of the TCSQ per experimental session, one per control scheme.

At the end of the test session, participants filled out the Control Scheme Preference Questionnaire. The study session took approximately 60 minutes to complete. Participants were given course credit or $10 (per their choice) as compensation for their time.

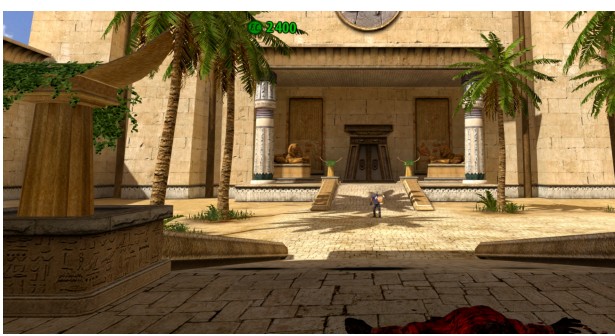

Figure 6: The Karnak Temple level, which participants played to acclimatize to VR.

### 3.4  Design

The experiment employed a 2 x 2 within-subjects design with the following independent variables (IV) and levels:

- **Control scheme**, with two levels: decoupled and coupled;

- **Task**, with two levels: locomotion and gaming context.

The locomotion task included two dependent variables (DV): (1) time (in seconds) and (2) number of errors. The time DV measured control scheme efficiency and was obtained by recording how long it took for the participant to traverse the single-path maze. The error DV measured control scheme effectiveness, an error occurred for each second the player's avatar touched a wall of the maze.

The gaming context task included two DV: (1) survival time (in seconds) and (2) number of kills. The survival time DV measured control scheme effectiveness and was obtained by recording how many seconds participants survived while playing the SS:TFE Bends on Sand level in "survival" mode. The kills DV measured control scheme effectiveness and represented the number of A.I.-controlled enemies killed by the participant.

To counterbalance possible order effects, we alternated the order of each control scheme for each participant. We always presented the locomotion task first, followed by the gaming context task.

To get a holistic view of the usability of the control schemes, we chose a convergent parallel mixed-methods approach where we gathered and analyzed quantitative and qualitative data and then merged and compared the results. Participants completed three questionnaires (1) the Task-Specific Control Scheme Questionnaire to gather qualitative and quantitative data on participant's control scheme experiences in that task context, (2) the Control Scheme Preferences Questionnaire to gather participants' impressions of the control schemes overall, not in the context of a specific task, and (3) the Immersive Experiences Questionnaire [Jennet et al. 2008]. See Appendix A.

### 4  RESULTS

We begin with the quantitative results related to usability, immersion, and control scheme preference. After the quantitative results, we present the qualitative analysis of the survey data. Our primary independent variable in both cases was control scheme.

### 4.1  Usability Results

Results are presented here for our two tasks, locomotion and gaming context.

#### 4.1.1  Locomotion Task: Errors

As shown in Figure 7, on average, participants made more errors with the decoupled control scheme (M = 5.15, SD = 5.66) than with the coupled scheme (M = 4.62, SD = 4.32). A paired samples t-test failed to detect a significant difference, t(25) = .82, p = .418, and the effect size was small, d = .11. Errors occurred when the player's avatar touched a wall for 1 second during the locomotion task.

#### 4.1.2  Locomotion Task: Time

As shown in Figure 7, on average, participants spent more time performing the locomotion task with the decoupled scheme (M = 40.58, SD = 17.74) than when using the coupled scheme (M = 38.31, SD = 14.41). However, a paired samples t-test did not find a significant difference, t(25) = -1.15, p = .263, and the effect size was small, d = .14.

#### 4.1.3  Gaming Task: Survival Time

Figure 7 depicts the average survival time across 4 trials. For this analysis, we used a two-way within-subjects ANOVA, applying the Greenhouse-Geisser adjustment when Mauchley's test of sphericity was significant. The ANOVA did not detect a

significant main effect of control scheme on mean survival time, F(1, 25) = .02, p = .879, $\eta$p2 < .01. There was also no significant main effect of trial on kills per control scheme, F(2.76, 69.09) = 2.02, p = .118, $\eta$p2 = .075. The analysis evaluated the control schemes' learnability and suggests little change in performance over time.

### 4.1.4 Gaming Task: Kills

Figure 7 depicts the mean kills in the gaming context task over 4 trials. Like survival time, an ANOVA found neither a significant main effect of control scheme, F(1, 25) = .16, p = .694, $\eta$p2 = .006, nor trial on kills, F(2.22, 55.61) = .63, p = .555, $\eta$p2 = .024, suggesting that participants' performance did not vary significantly over time.

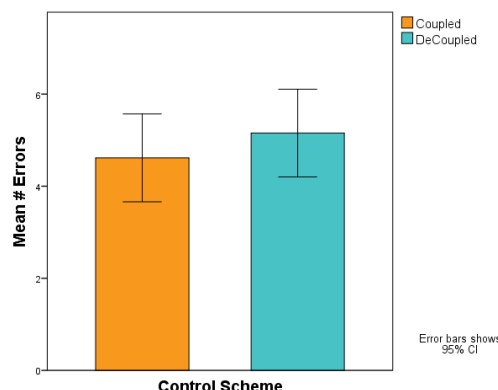

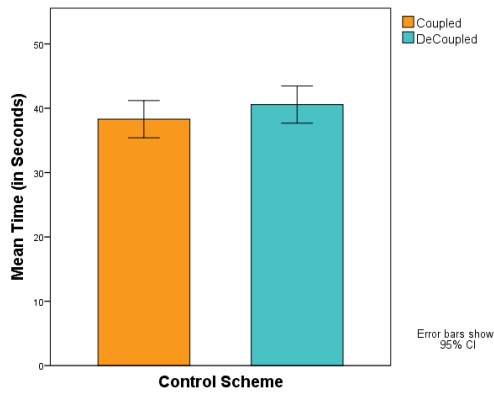

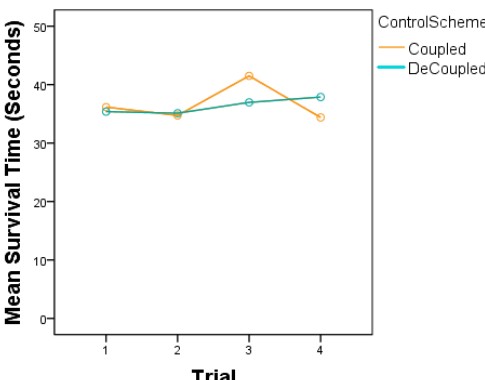

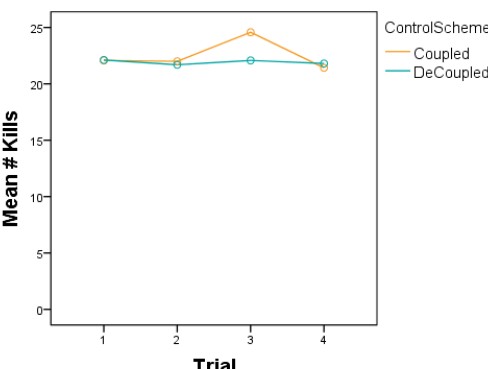

Figure 7: Usability Results. Locomotion task: mean time, errors by control scheme, Gaming task: mean survival time and kills by control scheme and trial..

## 4.2 Immersion Results

Recall that participants filled out the Immersive Experiences Questionnaire (IEQ) after playing the gaming context task for each control scheme. The questionnaire produced two immersion scores per participant: (1) an overall immersion score obtained by summing the ratings for the 31 questions in the IEQ, and (2) the "single question measure of immersion", in which the participant rated how immersed they felt on a scale of 1 to 10.

As shown in Figure 8, participants rated the decoupled control scheme (M = 143.42, SD = 11.91) as slightly more immersive than the coupled scheme (M = 140.46, SD = 11.43). However, a paired samples t-test did not find a significant difference, t(25) = -1.19, p = .247, and represented a small effect size, d = .25.

For the "single question measure of immersion", a paired samples t-test confirmed that there was no significant difference in immersion ratings for the coupled (M = 140.46, SD = 11.43) and decoupled (M = 143.42, SD = 11.91) control schemes, t(25) = -1.04, p = .247, d = .08.

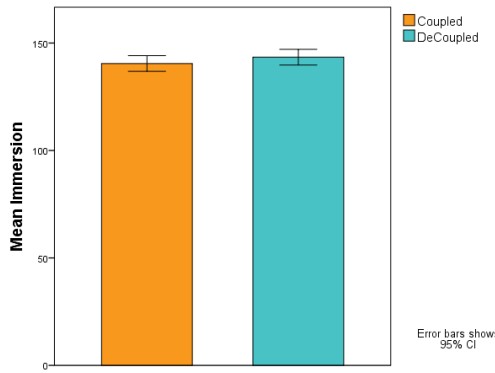

Figure 8: Mean immersion by control scheme type.

## 4.3 Preference Results

To determine which control scheme players preferred, we analyzed the control scheme rankings from the Control Scheme Preference Questionnaire to get a preference score for each control scheme type. As shown in Table , 14 participants preferred the coupled scheme and 12 preferred the decoupled scheme.

Rankings were compared using a Wilcoxon ranked-sign non-parametric test, which was not significant, $z = -.39$, $p = .841$.

We also asked participants to rate the naturalness and fun of each control scheme. The naturalness ratings for the coupled control scheme ($M = 6.88$, $SD = 2.40$) was slightly higher than for the decoupled control scheme ($M = 6.85$, $SD = 2.88$). A paired samples t-test did not find a significant difference in mean naturalness ratings, $t(25) = -.10$, $p = .918$, and the effect size was small, $d = .01$. While the fun ratings for the decoupled scheme ($M = 8.08$, $SD = 2.02$) was slightly higher than for the coupled scheme ($M = 8.04$, $SD = 1.78$) a paired samples t-test did not find a significant difference, $t(25) = .08$, $p = .936$, and the effect size was small, $d = .02$.

Table 1. Frequency table for control scheme rankings.

| Scheme | Ranked 1st | Ranked 2nd |
|---|---|---|
| Coupled | 14 (53.8%) | 12 (46.1%) |
| Decoupled | 12 (46.1%) | 14 (53.8%) |

## 4.4 Qualitative Results

To provide further insight into the results, we also obtained qualitative feedback from the participants on each control scheme.

### 4.4.1 Coupled Control Scheme Impressions

The feedback about the coupled control scheme was mostly favorable. Recall that in coupled control schemes, the HMD and right joystick are mapped to control the camera and steering the avatar. The participant can choose to use either input device.

The most common sentiments were that the coupled scheme was easy to use, natural, immersive, and accurate. Seven participants found the control scheme was easy to use, which was the most common feedback for the coupled scheme. Most of these participants indicated that they felt the coupled controls allowed them to determine what direction they will move in quickly. One participant said: "I felt more in control and I felt like I could react faster because moving my head controlled my direction". Five participants expressed a positive sentiment towards the coupling of the head and joystick control. Several liked the novelty of the interaction: "you have two different ways of controlling the way you move and it was cool having two ways.". Three participants also found head-based steering a practical choice, calling this interaction more accurate than using joystick. One participant noted that it "allowed for small adjustments when moving my head". Two found it more immersive, because of the body based "natural" movements, noting "I could use my actual face to move around the imaginary world which added to the feeling that I was part of it.". Four participants indicated that they found the coupled controls incorporated more natural movements into the control scheme, which affected the control of their character. One noted: "moving my head controlled my direction and I feel like that is a natural reflection."

While feedback was largely positive, four participants felt the control scheme was hard to control, finding the coupling of head and joystick control for steering and camera less usable because of confusion between head and hand input. One participant stated: "I wasn't always moving my head but when I was, I felt it was a little frustrating that I would always move in the direction that I'm looking. If checking a corner I would have to look, then turn my head back in order to move forward."

### 4.4.2 Decoupled Control Scheme Impressions

The feedback for the decoupled control scheme was not as positive as for the coupled scheme. While many participants found it easy to use, citing the scheme's familiarity, almost as many found it hard to use. Recall that the user could look around freely with the decoupled control scheme while steering their avatar with the right joystick.

The most common sentiment for the decoupled control scheme was that it was easy to use, with nine participants feeling this way. Many participants liked the separation of steering and camera control between different input devices, with the head controlling the camera and the right joystick controlling steering, indicating that it "filters out the confusion of having directional movement with both looking and joystick.". Seven participants found using the right joystick for steering without interference from head movements (as in the coupled scheme) a familiar experience. One said, "I felt that it was a little simpler and straight forward. Perhaps that is because I grew up relying on controllers for in-game movement, rather than swiveling my head.". Two participants noted that head movement did not control steering, which was an advantage for detecting threats in the game world. One participant said, "I rely on the controllers to move around while having the freedom to use my head, to scan the field of view."

Conversely, an equal number of participants found the decoupled control scheme hard to control. The most common comment was that the participant's awareness of their avatar's facing direction would get out of sync with their head direction, causing them to have to re-calibrate. One participant said, "If there was a way to see where my body was facing (like seeing my legs for example) I think I would have preferred this control scheme. The main drawback was forgetting where the front of my body was facing if I turned my head around.". There were also some easily-corrected usability issues with this scheme. Five participants found the right joystick's sensitivity was too high, which may have affected the participant's judgment, although the sensitivity can be modified by the user in the options menu.

As a decoupled control scheme, players had to use head movement to move the camera view. Two participants mentioned this as an issue, specifically when trying to look up or down. As one participant explained, "My biggest problem with this scheme was that I couldn't look up and down using my controller (like I would normally be able to do using console controllers); it felt like I was constrained to some kind of horizontal movement path where the only way to look up was using my actual face. Changing that would give the player MUCH more control over his character." If our maps required shooting enemies above or below the horizon line, it is possible that the decoupled control scheme would affect participants' performance in the gaming context task.

## 5 DISCUSSION

In this study, we compared coupled and a decoupled control schemes in a room-scale VR game setting using Oculus Touch motion controllers, Oculus Rift HMD and position sensors as input devices. We hypothesized that the coupled control scheme would be more usable, immersive and preferred by players over the decoupled scheme for room-scale FPS VR games, based on the results of studies for stationary VR games [Martel and Muldner 2017]. However, the quantitative results found no significant effect of control scheme on our usability metrics: efficiency, effectiveness, satisfaction and learnability. The "kills" results may have been due to both control schemes using the

Oculus Touch motion controllers to target, which were decoupled from camera view. Other studies have shown significant difference [Martel and Muldner 2017] between control schemes for targeting tasks (e.g., studies that mapped the more precise mouse input to targeting for the coupled control scheme, whereas the decoupled scheme mapped the HMD to targeting).

Player behavior may have also affected the error and time results for the locomotion task. Recall that participants could use either the HMD or the right joystick to steer their avatar in the coupled scheme. One possibility is that participants who used head steering for the coupled scheme may have increased error rates during the locomotion task compared to using the right joystick. This fact highlights the importance of tracking participants' behavior to ascertain which input device they actually used and is a limitation of the this research.

Nevertheless, the qualitative results revealed a more complex picture. While the coupled scheme had overwhelmingly positive feedback, the participants reported usability issues with the decoupled control scheme. The most common issue was that the separation of the avatars movement direction and camera view confused participants and was therefore inefficient, a result other studies have also found [Martel and Muldner 2017]. The usability issues inherent in the decoupled control scheme did not prevent approximately half (12) of the participants from preferring that scheme (whereas just over half, i.e., 14, participants preferred the coupled scheme overall). Although feedback for the coupled scheme was largely positive, four participants found the coupling of the head and joystick input confusing.

We hypothesized that the coupled scheme would be more immersive. However, our hypothesis was not confirmed in either the quantitative immersion scores (which did not show a significant effect of control scheme on immersion) or qualitative feedback from participants.

We tested the control schemes in two contexts, as prior work has shown that control schemes can perform differently for different VR tasks. We chose a locomotion task, where the player's agility using the control schemes was the focus, and an ecologically valid gaming context task where the player had to fight and defend against A.I.-controlled enemies. We hypothesized that the coupled control scheme would have better usability scores regardless of task context, but there were no significant differences between the two control schemes for the quantitative usability metrics for either task.

Our goal was not only to measure usability, but to generate best-practice recommendations for control scheme design. Based on our analysis of qualitative feedback about the two control schemes, we identified two control scheme factors that contributed to usability issues in controller-based room-scale VR games. These usability factors can be used by game designers when choosing a control scheme or creating novel control schemes. The usability factors and their implications are as follows:

**Coupled vs. decoupled controls.** Although the quantitative results for the usability, preference and immersion metrics were similar between the two control schemes, our qualitative analysis revealed usability issues with the decoupled scheme. Our analysis suggests that overall, the most important factor affecting player sentiment was whether the control scheme was coupled or decoupled. The feedback for the coupled scheme was almost wholly positive, while the feedback for the decoupled scheme was mixed, with most of the negative feedback centering on the decoupled nature of the controls. With the decoupled control scheme, the camera was controlled by the HMD, while steering was controlled by the right joystick. whereas with the coupled scheme camera and steering are controlled by both the joystick and HMD. When using the decoupled scheme participants had difficulty remembering which direction their avatar was facing (controlled by the right joystick) relative to the direction their head (and the camera) was facing. When their awareness of their avatar's facing direction got out of sync with their head direction, players had to stop and think for a moment to reorient themselves, which they found inefficient. While a few participants had the opposite opinion and disliked coupled controls because head movement could unintentionally interfere with steering their avatar; overall, the coupled scheme had more consistently positive feedback. Therefore, we recommend it as the default control scheme for action-oriented, room-scale, controller-based VR games. Because many participants enjoyed the decoupled scheme, and four participants found the coupled controls hard to use, we recommend including decoupled control scheme as an alternative to the default coupled control scheme. Game designers may also find it worthwhile to allow players to "re-sync" the avatar and camera direction while playing. For example, by pressing the right joystick or another controller button, the avatar facing direction could be automatically set to the same rotation as the camera.

**Input device degrees of freedom.** When using the decoupled control scheme, participants felt like their joystick input was constrained when not being able to use the right joystick to look up or down, forcing them to use head movement. While the camera view is supposed to be controlled by head movement in decoupled schemes, it might be more effective if both the HMD and right joystick are mapped to rotating the camera around the x-axis. Decoupled controls could dramatically impact usability in environments that position enemies above or below the player's eye level as it would require participants to use head movement to aim at targets, which prior research has found is less accurate than mouse or joystick input [Gerling et al 2011; Klochek and MacKenzie 2006; Natapov et al 2009]. Previous work that suggests constraining DOFs yield lower performance on targeting tasks and affects player preference also supports this result [Martel and Muldner 2017].

The summary above shows that while there was no significant main effect of control scheme on usability and immersion metrics, the qualitative results tell a different story. While the decoupled scheme was preferred by 12 of the 26 participants, the inherent decoupled-ness of the scheme caused usability issues for some, as it was difficult for some participants to maintain awareness of their avatar's facing direction. The coupled scheme was the most usable as judged by more consistently positive feedback and fewer major usability issues reported for this scheme. These findings highlight the importance of gathering qualitative user feedback data in addition to quantitative usability metrics.

## 6 LIMITATIONS AND FUTURE WORK

In this study, we evaluated the usability of coupled and decoupled control schemes for controller-based locomotion in room-scale VR FPS games. A limitation of this work is the lack of control we had in specifying the values for the control scheme's control-display relationships. Programmers at Croteam may have used different values for transfer functions, implemented dead zones differently between the two schemes or any other differences that could affect usability. Player feedback suggests that joystick sensitivity for the coupled scheme was more pronounced than the decoupled scheme, for example. A more controlled comparison

between schemes may elicit stronger differences in quantitative measures.

Another limitation of this work is that the results may not be generalize to VR games that use other input devices or are stationary as opposed to room-scale. For example, it is possible that that the position tracking of room-scale VR may allow users to make adjustments to their position and rotation using body pose when playing. This could conceivably make the decoupled control scheme more usable, as some prior research has shown that real-walking is more effective and efficient for locomotion and navigation tasks [Suma et al. 2007; Langbehn et al. 2018; Buttussi and Chittaro 2021].

While our study is a first step towards evaluating control schemes for FPS VR games, future work could include research into how to alleviate the usability issues inherent in decoupled control schemes. Inventing and evaluating methods for helping players maintain awareness of their avatar's facing direction could help make the decoupled scheme more usable. Participants also found it awkward to look up or down using head movement when using the decoupled control scheme. Mapping the right joystick to rotate the camera around the x-axis, while also letting the HMD control the camera, may be a welcome addition to the decoupled control scheme mappings.

Our study focused on FPS gameplay in particular. Because task context affects the performance of a control scheme, our results likely do not generalize to all VR games, and thus further study would be needed for control schemes for other game genres and game POV other than first-person. Future directions could also include testing a more diverse set of VR control schemes. Although two categories of schemes (the coupled and decoupled schemes) are among the most commonly found in present day VR games, other control schemes may replace them as VR games mature. Finally, this research cannot be generalized to novice gamers. Experienced gamers are accustomed to using the mouselook control scheme and may be biased in favor of similar control schemes. It would be useful to discover the control scheme preferences in populations without pre-existing biases.

## ACKNOWLEDGMENTS

Acknowledgements.

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

Summary of measures and data analysis methods used in the mixed-methods study.

| Data Type | Variable | Measures Used | Collection Method | Data Analysis |
|---|---|---|---|---|
| QUAN | Effectiveness | Performance scores | Locomotion Task: Error rate (Derived from number of collisions with walls)

Gaming Task: Enemies killed ("kills") and survival time. | Paired t-test |
| | Efficiency | Performance scores | Locomotion Task: Time to complete maze | Paired t-test |
| | Learnability | Performance scores | Gaming Task: Track effectiveness and efficiency scores over 4 trials. | 2-way ANOVA with control scheme and trial as IV |
| | Satisfaction | *Control Scheme Preference Questionnaire* | "Preference" Score was derived from question in CS Preference Questionnaire:
Q. Please rank the control schemes from the most preferred to least preferred.
"Fun" score:
Q. How natural did it feel to use the control scheme?
(Not natural 1 2 3 4 5 6 7 8 9 10 Very natural)
"Naturalness" score:
Q. How fun was it to use the control scheme?
(Not at all fun 1 2 3 4 5 6 7 8 9 10 Very fun) | Preference score: Wilcoxon rank.

Fun and Naturalness scores: paired t-tests |
| | Immersion | *Immersive Experiences Questionnaire* | Likert rating scale | Paired t-test |
| QUAL | Satisfaction | *Task-Specific Control Scheme,*
and;
*Control Scheme Preference Questionnaires* | The *Task-Specific Control Scheme Questionnaire* gathered feedback about each control scheme for the locomotion task:
Q. What did you like about using the coupled/decoupled control scheme for the maze map?
Q. What did you dislike about using the coupled/decoupled control scheme for the maze map?
*The Control Scheme Preference Questionnaire* probed for reasons behind control scheme preference rankings:
Q. Why do you prefer the control scheme you ranked first more than the other?
Q. Why do you like the control scheme you ranked last the least?
Q. If you have any other comments about any of the control schemes used in this study, please write them below. | Qualitative analysis: Identify participant feedback relating to satisfaction. |
| | Immersion | *Task-Specific Control Scheme Questionnaire;*
*Control Scheme Preferences Questionnaire* | (see cell above for relevant questions) | Qualitative analysis: Identify participant feedback relating to immersion. |