# OpenReview forum: "Control Schemes for Room-Scale VR Games"
_graphicsinterface.org/Graphics_Interface/2023/Conference — Submitted to GI 2023_

### Official Review · Reviewer_n5sf · 2023-01-13
**Paper Review**

**Rating:** 2
**Confidence:** 5

**Review:**

The focus of this paper is to compare different navigation control schemes in the context of room-scale VR games.  The first control scheme was gaze directed and the second was hand directed.  These schemes were compared in two different interaction tasks and both quantitative data and qualitative data was collected. Statistical tests were performed on the quantitative data and no significant differences were found.  The qualitative data showed that some participants found the hand directed control scheme to be problematic in that it was difficult to maintain awareness of the avatar's facing direction.  The paper also highlights the importance of gathering qualitative data in addition to quantitative user metrics.

I commend the authors for the work they put into this paper and the work is certainly asking an interesting question.  The paper is clear and easy to read and understand and the references seem appropriate although the paper should probably cite

LaViola, J., Kruijff, E., McMahan, R., Bowman, D., and Poupyrev, I. 3D User Interfaces: Theory and Practice, Second Edition, Addison Wesley, ISBN 0134034325, April 2017

as a lot of what the paper states is in this text.  Unfortunately, the results from this paper are not significant and it does not make a contribution to the field.  The importance of gathering qualitative data in addition to quantitative data is already known. It would be good for the authors to explore other navigation methods, perhaps more modern ones, to see how they affect user experience. As it stands this paper is not ready for GI.

---

### Official Review · Reviewer_D4rp · 2023-01-13
**The paper presents a small scale study to compare coupled and decoupled control in room-scale VR games. The results show that there is little difference in quantitative measures, but somewhat more pronounced difference in the opinions of the study participants in favour of coupled control.**

**Rating:** 7
**Confidence:** 4

**Review:**

The paper is well written. The study is described clearly, and the results sufficiently discussed. This paper could be of interest to the audience of GI 2023 that is interested in room -VR applications.

The study is somewhat limited, involving only 26 people, between the ages of 18-30. Most of the participants had prior experience with games, and 15 of them have used VR headsets. The two experimental setups (FPS, Maze) and the dependent variables for the quantitative analysis, were reasonable. I would expect that coupled control is more suitable for FPS rather than a game where the user walks around looking for objects and clues, so  it is surprising to me that this is not reflected in the quantitative results.

It was not clear to me how large the room was, or how much the participants had to walk in the FPS case.

A larger scale study that include additional experimental set-ups would be really useful.

Overall the paper is informative and interesting and could be published in GI 2023.

---

### Official Review · Reviewer_ySKD · 2023-01-20
**The paper presents a study investigating the impact of coupling gaze direction with motion control in head-mounted display-based VR systems.**

**Rating:** 6
**Confidence:** 4

**Review:**

The paper presents a study investigating the impact of coupling gaze direction with motion control in head-mounted display-based VR systems. The study uses the Oculus Rift CV1 and seeks to understand the impact of coupling gaze direction on usability, immersion, the interaction between task and usability, and familiarity with existing game controls. The claim is that prior works have only focused on seated VR in their analysis and that the presented study considers room-scale games where participants are free to move within a local area and also navigate beyond the physical boundaries of the room via the navigation control scheme.

The clarity of exposition Is very good. The paper is easy to read and understand.

The quality is good. From a technical standpoint, the study and analysis are detailed enough to be repeatable (save for environmental conditions details like scale).

The originality is perhaps difficult to judge. There is a large number of works in VR navigation and VR interaction techniques. Some of the related works are light on details for such mature areas. It seems the very controlled study is original in the small set of questions it seeks to answer while focusing on gaze couple navigation and the subset of that area useful for games. However, I suspect the many VR navigation papers address similar questions.

Related to originality the significance of the work is not high. There are no strong takeaways from the paper. The question of gaze coupled versus non-gaze coupled (the ability to look around versus moving in the direction of looking) is perhaps not so strong, in that you can look still in not gaze coupled if you are not using the movement control. So really the gaze degrees of freedom are only lost while moving. Both tasks in the study do not necessitate this, you can complete them in steps not required while moving.

Section 2.1 on VR Interaction Techniques is largely under-cited and dated focusing on mainly a single paper from 1999 when this an active area of research related to the work.

Similarly, section 2.2 Game immersion, virtual reality immersion, judgement\perception of space is a large field with active research and this section is rather dated and under-cited. The focus mainly seems to be providing reasoning for a methodological choice—which questionnaire to use, rather than works in the related space.

From a methodology standpoint, focusing on coupling gaze or not makes for a cleaner study of control schemes where there are numerous options. The framing of the paper seems much broader though. For example, the title does not reflect the scope of the paper, which is more akin to “The impact of gaze coupled control schemes in room-scale VR”

The methodology seems clear enough to repeat. However, I would argue that the details on the locomotion map condition are sparse. It would seem the scale of the environment would be important. Figure 5 would be better represented orthographically, or even better as a blueprint with clear size indications.

There are no quantitatively significant results across usability, immersion, or preference. I suspect a deeper study with more participants is needed to understand what is going on and whether the form of gaze coupling studied has the impact the authors hoped for.

The paper's impact seems to rest heavily on the qualitative results, section 4.4. However, with a small participant pool and few or no salient results (most feedback is given by only singular participants).

---

### Meta-Review · Area_Chair_gWwC · 2023-01-23

**Recommendation:** 5
**Confidence:** 5

**Metareview:**

The reviewers all found the paper to be an interesting research area and timely topic.  Reviews were fairly mixed with 2 reviews scoring the paper as above the bar for GI and one review clearly below the bar for GI.  From the reviews, the pros for the paper include:

+ clarity of exposition is good
+ easy to understand
+ study clearly described and results sufficiently discussed
+ work clear enough to repeat

The cons for the paper include:

- no statistically significant findings
- no strong takeaways from the findings
- study is limited, lacking in number of participants

Given the split reviews, the cons seem to outweigh the positives with this paper, making it just below the acceptance threshold for GI.